# Efficient Linear System Solver with Transformers

**Max Vladymyrov** [1]   **Johannes von Oswald** [1]   **Nolan Miller** [1]   **Mark Sandler** [1]

## Abstract

This paper investigates the potential of linear Transformers as solvers for systems of linear equations. We propose a novel approach where the Transformer encodes each equation as a separate token, allowing the model to process the system in a permutation-invariant manner. To enhance generalizability and reduce the parameter count, we introduce a block-wise re-parameterization technique for the attention weight matrices. This technique decouples the problem dimension from the model's parameter count, enabling the Transformer to effectively handle systems of varying sizes. Our experiments demonstrate the Transformer's competitive performance compared to established classical methods such as Conjugate Gradient, especially for systems with smaller sizes. We further explore the model's ability to extrapolate to larger systems, providing evidence for its potential as a versatile and efficient solver for linear equations.

## 1. Introduction

Solving linear systems of equations is a fundamental problem in numerous fields, including scientific computing, machine learning, and engineering. While traditional methods like Gaussian elimination and iterative solvers like Conjugate Gradient (Hestenes et al., 1952) are widely used, exploring alternative approaches holds the potential for more efficient and versatile solutions.

Transformers, originally developed for natural language processing tasks (Vaswani et al., 2017), have demonstrated a remarkable ability to capture complex relationships within sequential data. This ability extends beyond natural language processing, as evidenced by their successful application in solving a variety of problems, ranging from noisy linear regression and classification (Garg et al., 2022) to the traveling salesmen problem (Yang et al., 2023) and other domains (Mirchandani et al., 2023). Transformers have shown promise in various scientific computing tasks beyond natural language processing (Li et al., 2020).

In this paper, we investigate the ability of Transformers to solve linear systems of equations of the form $Ax = b$, where $A$ is a positive definite and symmetric matrix. This approach presents several compelling advantages:

- Potential for Parallelism: Transformers' inherent parallelism in processing tokens could translate to efficient solvers, particularly for large-scale systems.

- Extrapolation Capabilities: We explore the potential of Transformers to generalize beyond the size they are trained on, offering a more flexible tool.

We propose a novel tokenization scheme that encodes each equation as a separate token, allowing the Transformer to process the system in a permutation-invariant manner. We demonstrate that even linear Transformers can solve linear systems of equations with high accuracy.

Since the attention does not have memory, the whole linear system must be provided in-context. A naive encoding scheme would result in the number of model parameters scaling quadratically with the size of the linear system. To address this, we introduce a novel re-parameterization technique for the attention weight matrices that decouples the problem dimension from the model's parameter count. We empirically demonstrate that this re-parametrization does not compromise accuracy substantially while significantly reducing the number of parameters, requiring only dozens of parameters per layer. This approach enables the Transformer to effectively handle systems of varying sizes.

Our experiments demonstrate that the proposed Transformer-based solver achieves comparable accuracy to 6-8 iterations of the Conjugate Gradient method, while demonstrating superior speed for systems with a small number of equations. Furthermore, we investigate the model's ability to extrapolate to larger systems, showcasing its potential as a versatile and efficient solver for linear equations. This research contributes to the understanding of Transformers' capabilities beyond traditional sequence processing tasks and opens avenues for further exploration

---

[1]Google Research. Correspondence to: Max Vladymyrov <mxv@google.com>.

*The first AI for MATH Workshop at the 41st International Conference on Machine Learning*, Vienna, Austria.

of their applications in numerical problems.

## 2. Preliminaries

In this section we introduce notations for linear Transformers, data representation, and the specific type of problem we consider.

### 2.1. Linear Transformers

Given an input sequence $e_1, e_2, ..., e_N \in \mathbb{R}^D$, a single head in a linear self-attention layer is typically parameterized by four matrices, key $W_K$, query $W_Q$, value $W_V$ and projection $W_P$. The output of the non-causal layer at position $i$ is $e_i + \Delta e_i$ where $\Delta e_i$ is computed as

$$\Delta e_i = W_P \left( \sum_{j=1}^{N} \langle W_Q e_i, W_K e_j \rangle W_V e_j \right). \qquad (1)$$

Equivalently, we can use the parameters $P = W_P W_V$ and $Q = W_K^\top W_Q$, resulting in the equation:

$$\Delta e_i = \sum_{j=1}^{N} (e_j^\top Q e_i) P e_j. \qquad (2)$$

In the case of multiple heads $(P_1, Q_1), (P_2, Q_2), ..., (P_h, Q_h)$, the effect is simply the summation of all heads:

$$\Delta e_i = \sum_{k=1}^{H} \sum_{j=1}^{N} (e_j^\top Q_k e_i) P_k e_j. \qquad (3)$$

We define a *linear Transformer* as a multi-layer neural network composed of $L$ linear self-attention layers parameterized by $\theta = \{Q_k^l, P_k^l\}_{k=1...H, l=1...L}$. To isolate the core mechanisms, we consider a simplified decoder-only architecture, excluding MLPs and LayerNorm components. This architecture has been used in previous work (von Oswald et al., 2023; Ahn et al., 2023; Vladymyrov et al., 2024). For clarity, we focus on the single-head attention case and drop the index $k$.

### 2.2. System of linear equations

Our goal is to find a vector $x \in \mathbb{R}^N$ that solves the system of $N$ linear equations $\langle a_i, x \rangle = b_i$ for each $i = 1 \ldots N$ with $a_i \in \mathbb{R}^N$, and $b_i \in \mathbb{R}$. In matrix form, this problem can be written as $Ax = b$, where $A \in \mathbb{R}^{N \times N}$ and $b \in \mathbb{R}^N$.

We focus on the case where $A$ is a positive definite symmetric matrix with a fixed condition number $\kappa$. The condition number significantly impacts the convergence of iterative methods (Trefethen & Bau, 2022). In this paper, we present preliminary results with $\kappa = 5$.

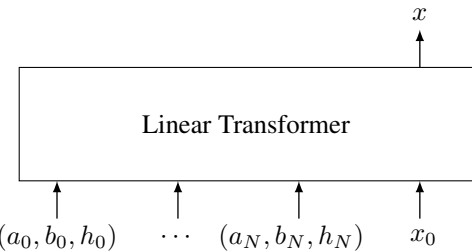

*Figure 1.* A Transformer receives an input embedding where each token corresponds to the parameters $(a_i, , b_i)$ of a single equation within a linear system. A learned or predefined embedding $h_i$ can be added for improved performance. The initial solution estimate $x_0$ is fed in as the final query token, and the final solution $x$ is obtained after passing through the Transformer layers.

### 2.3. Encoding of the tokens

The Transformer processes input data as a sequence of tokens. Each equation in the linear system is represented as a distinct token $e_i = (a_i, b_i, h_i) \in \mathbb{R}^D$, where $D = N + 1 + K$ and $h_i \in \mathbb{R}^K$ are optional embedding vectors. These embeddings can be either pre-defined or learned during model training. We can represent them collectively in a matrix form as $H \in \mathbb{R}^{N \times K}$. This tokenization scheme ensures that the model remains invariant to the order in which the equations are presented.

Additionally, we append a query token $e_{N+1} = (x_0, 1_{1+K})$ to the sequence, where $x_0 \in \mathbb{R}^d$ is an input placeholder from which the output will be read, initialized with all zeros. We constrain the attention mechanism to focus solely on the first $N$ tokens of the sequence, ignoring the query token.

We use superscript notation (e.g. $a_i^l$, $b_i^l$) to denote the corresponding component of the $i$-th token in the Transformer's output at layer $l$. The initial layer corresponds to the input: $(a_i^0, b_i^0) = (a_i, b_i)$. For a model with parameters $\theta$, the prediction is obtained as a transformation of the $x_0$ component of the final token in the last layer: $f_\theta(\{e_1, ..., e_N\}, e_{N+1}) = x_{N+1}^L$.

To train the linear Transformer using the tokens, we minimize the following mean squared error loss function using batches of different generated problems $\{A, b, x\}$:

$$L(\theta) = \mathop{\mathbb{E}}_{A,b} \left[ (f_\theta(\{e_1, ..., e_N\}, e_{N+1}) - x)^2 \right]. \qquad (4)$$

## 3. Re-parameterization of weight matrix

Given that each token consists of three components $A$, $b$ and $H$, we can equivalently rewrite the attention weight matrices by expanding the interaction between each component:

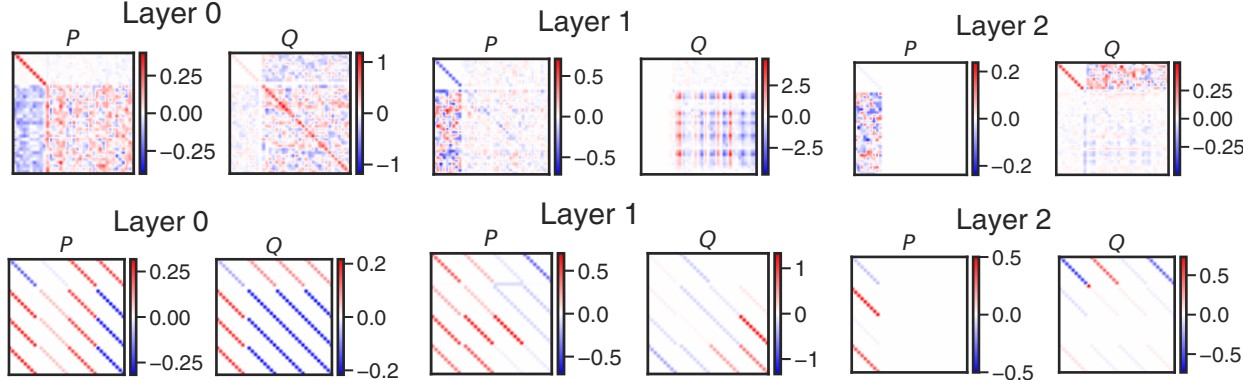

*Figure 2.* Learned weights for the 3-layer linear attention for $9 \times 9$ linear system solver. *Top:* FULL encoding with a 27-dimensional extra embedding $h_i$. *Bottom:* BLOCK encoding with 3 $N \times N$ identity matrices as extra embeddings.

$$P = \begin{pmatrix} P_{A,A} & P_{A,b} & P_{A,H} \\ P_{b,A} & P_{b,b} & P_{b,H} \\ P_{H,A} & P_{H,b} & P_{H,H} \end{pmatrix},$$

$$Q = \begin{pmatrix} Q_{A,A} & Q_{A,b} & Q_{A,H} \\ Q_{b,A} & Q_{b,b} & Q_{b,H} \\ Q_{H,A} & Q_{H,b} & Q_{H,H} \end{pmatrix}.$$

Assuming for simplicity that the dimension $K$ of vectors $h_i$ is a multiple of $N$ (i.e. $K = lN$), $H$ can be decomposed into $l$ blocks $H^l \in \mathbb{R}^{N \times N}$. Consequently, the components of $P$ and $Q$ controlling the interaction between $A$ and all the $H^l$ will have a size $N \times N$. Components handling the interaction with $b$ will have sizes $N \times 1$, $1 \times N$ or $1 \times 1$.

We propose the following block-wise re-parametrization the attention weights matrices. Each rectangular matrix is represented as $\mu 1_{N \times 1}$ or $\mu 1_{N \times 1}^{\top}$, while each square matrix is represented as $\mu_1 I_{N \times N} + \mu_2 1_{N \times N}$. Here $I$ is the identity matrix, and $1$ is a matrix where all elements are ones.

Our approach draws inspiration from low-rank approximations in numerical linear algebra (Halko et al., 2011) and is motivated by several factors:

- Columns independence of $A$. The entries of $A$ are sampled independently, preventing the Transformer from favoring any specific dimension. With sufficient training data, the Transformer learns to apply the same weight to the first $N$ dimensions (corresponding to $A$) and a separate weight to $b$.

- Performance. Our experiments show that training the model with this parametrization achieves a loss comparable to training all weights independently.

- Efficiency. utilizing a single scalar for each block significantly reduces per-block computation, enhancing the overall speed of the algorithm.

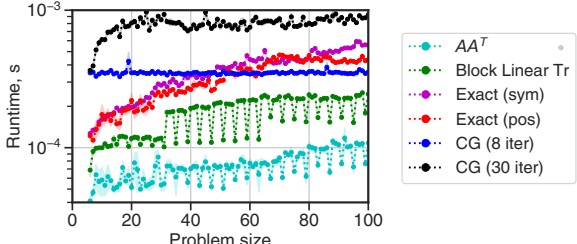

*Figure 3.* Runtime comparison of different linear system solvers on an A100 GPU, using batches of $1\,000$ systems with varying matrix sizes.. $AA^T$ represents the runtime for a batch of $1\,000$ matrix multiplications of matrix $A$ provided for reference. BLOCK LINEAR TR is our proposed algorithm with 3 layers of linear Transformer using BLOCK encoding. EXACT (SYM) and EXACT (POS) are computed using `jax.scipy.sparse.linalg.cg()` solver assuming symmetric of positive definite matrix structure for $A$, respectively. CG is computed using `jax.scipy.sparse.linalg.cg()` solver for a given number of iterations.

- Generalization. This method decouples the problem dimension $N$ from the number of model parameters. This allows us to apply the learned model to inputs of different sizes, not just the size it was trained on. We can also fine-tune model trained on one size for another.

We refer to this new re-parameterization as BLOCK embedding, as opposed to FULL embedding where the full weight matrices are learned. Figure 2 shows the learned weights of the trained FULL and BLOCK embeddings for a 3-layer linear Transformer.

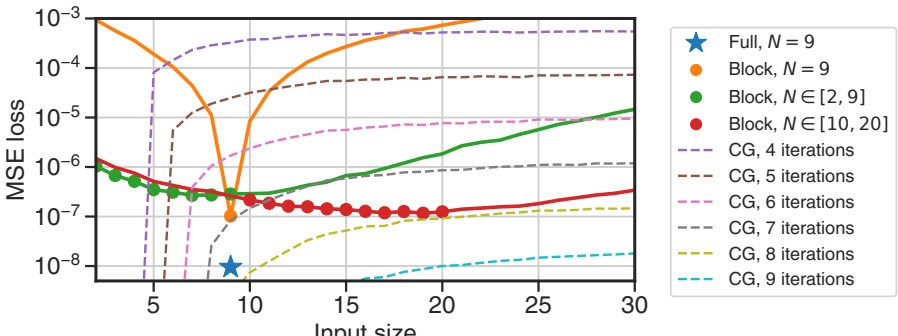

*Figure 4.* MSE loss comparison of several 3-layers Transformer-based linear system solvers against a Conjugate Gradient baseline with varying numbers of iterations. Markers indicate the problem size where the method is applied within its training domain. Solid lines indicate out-of-domain generalization. The dashed line represents the baseline performance.

## 4. Experiments

Figure 3 compares the runtime required to compute the solution using different solvers on A100 GPU. Our proposed method (BLOCK LINEAR TR) requires only a few matrix multiplications to compute a forward pass through 3 layers of the linear Transformer. While the Conjugate Gradient (CG) method theoretically scales as $\mathcal{O}(N^2)$, which is asymptotically faster than matrix multiplication's $\mathcal{O}(N^{2.8})$, the constant overhead associated with CG becomes significant for the small values of $N$ considered here. Moreover, matrix multiplication is a more native operation for GPUs compared to the CG, which involves dot products and matrix-vector products. In this context, computing 3 layers of linear attention proves to be considerably faster than competing methods.

Figure 4 compares the accuracy of our solver. We trained several variants of the 3-layer linear Transformer:

- FULL, $N = 9$. This variant trains full weight matrices on problems with size $N = 9$ only. Due to the fixed $9 \times 9$ weight dimensions, this model cannot generalize to matrices of other sizes. However, it achieves the best performance we observed for problems of size $N = 9$ using linear Transformers.

- BLOCK, $N = 9$. This variant trains weights using BLOCK encoding with 3 $N \times N$ identity matrices as extra embeddings, also on $N = 9$ problems only. While the results can be applicable to problems of other sizes, the generalization quality is limited.

- BLOCK, $N \in [2, 9]$. This variant is trained with BLOCK encoding for problems with sizes $N \in [2, 9]$. It performs willwithin this range and exhibits generalization to other sizes.

- BLOCK, $N \in [10, 20]$. This model was fine-tuned from model above trained on $N \in [2, 9]$ using data

with sizes $N \in [10, 20]$. The performance significantly improves the sizes $N \in [10, 20]$, albeit with a small degradation in performance for sizes $N \in [2, 9]$.

Compared to the Conjugate Gradient baseline, our algorithm achieves an accuracy roughly equivalent to 6-8 iterations of CG, depending on the input size.

## 5. Conclusions

This paper explored the novel application of linear Transformers for efficiently solving small systems of linear equations with symmetric and positive definite coefficient matrices. We demonstrated that by encoding each equation as a distinct token and implementing a block-wise reparameterization technique, Transformers could achieve accuracy comparable to 6-8 iterations of Conjugate Gradient, while being faster for small problem sizes. Our model exhibited the ability to generalize beyond its training data, effectively handling systems of varying sizes and unseen condition numbers.

This research opens up exciting possibilities for utilizing Transformers in numerical tasks. Further investigation into architectural choices, training strategies, and the boundaries of generalization could lead to the development of even more efficient and adaptable solvers. This work reinforces the notion that Transformers, initially designed for natural language processing, hold remarkable potential as powerful tools across various scientific computing domains.

While our approach shows promise, it has several limitations. Currently, we only handle positive definite symmetric matrices, which restricts the applicability of our method. Future work should explore extending this approach to general matrices, including non-symmetric and indefinite cases. Additionally, our method's performance on very large systems or highly ill-conditioned matrices needs further investigation.

Scaling the approach to handle sparse matrices efficiently is another important direction for future research. Integrating our Transformer-based solver with classical methods, potentially as a preconditioner or in a hybrid algorithm, could leverage the strengths of both approaches and is an exciting area for further study.

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
