# OpenReview forum: "Efficient Linear System Solver with Transformers"
_ICML.cc/2024/Workshop/AI4MATH — ICML 2024 Workshop AI4MATH Poster_

### Official Review · Reviewer_dVvN · 2024-06-10

**Rating:** 5
**Confidence:** 3

**Summary:**

This paper articulates a novel linear transformer architecture to solve linear systems. Equations are encoded as permutation-invariant tokens and re-parameterized to decouple the problem dimension from the model’s parameter count. Experiments show that the proposed method has a competitive performance against the SOTA linear solvers.

**Questions:**

See "Reasons To Reject" above. I would recommend an acceptance after I see additional experimental data on the issues detailed above.

**Reasons To Accept:**

- This paper has enough novelty. Transformer-like architecture seems like a good choice for fighting varying problem dimensions.

**Reasons To Reject:**

- Experiments are hand-waving. Only small-scale problems (up to a problem size of 100) are tested, while numerical solvers typically need to deal with large-scale problems (say, $Ax = b$ where $A$ is a sparse matrix of size 1000000x1000000).
- The authors claim a "competitive performance", yet no quantitative results are reported in the experiments section (e.g., time cost, final accuracy achieved).
- The baseline is not convincing. In practice, CG solvers are typically preconditioned with multigrid (MG) preconditioners. A sole CG iterator does not fully represent the SOTA.
- The authors evaluate the MSE error during training and testing. This is OK for intermediate steps, however, in practice, the final precision of a numerical solution is evaluated by the relative residual of the solution, $r(x) = ||Ax - b|| / ||b||$. I would like to see the quantitative data on residual errors.
- This manuscript is formatted with the NeurIPS 2024 template, please re-format it with the ICML template. Also, is this manuscript also submitted to NeurIPS 2024?

---

### Official Review · Reviewer_tFdu · 2024-06-11

**Rating:** 4
**Confidence:** 4

**Summary:**

This paper studies the application of linear Transformers for solving a small systems of linear equations with symmetric and positive definite coefficient matrices. To reduce the number of parameters, a novel re-parameterization technique for the attention weight matrices is introduced. While this is an interesting research, its applicability in practice is questionable.

**Questions:**

More discussion on figure 2 should be added. In a given layer, the structures in top and bottom are quite different. What are the main points that the authors are trying to state with figure 2?

**Reasons To Accept:**

This work is very relevant and of broad interest

**Reasons To Reject:**

1. The literature review is very limited. In AI/ML literature, there are several recent works on applying Transformer for solving/studying dynamical systems and linear systems. For instance:

@article{geneva2022transformers,
  title={Transformers for modeling physical systems},
  author={Geneva, Nicholas and Zabaras, Nicholas},
  journal={Neural Networks},
  volume={146},
  pages={272--289},
  year={2022},
  publisher={Elsevier}
}

@article{charton2021linear,
  title={Linear algebra with transformers},
  author={Charton, Fran{\c{c}}ois},
  journal={arXiv preprint arXiv:2112.01898},
  year={2021}
}

@incollection{arslan2021machine,
  title={Machine Learning Algorithms for Solving Linear Systems of Equations},
  author={Arslan, Hilal and Bozyigit, Fatma},
  booktitle={Developing Mathematical Literacy in the Context of the Fourth Industrial Revolution},
  pages={155--184},
  year={2021},
  publisher={IGI Global}
}
Also, there are also works on other machine learning methods to solve linear system in the AI/ML literature. However, the author did not review/compare these works in the same field/direction.

2. The representation of this paper can be improved. For instance, line 109: 'We can this new re-parametrization BLOCK embedding...'
3. Other machine learning-based approach for solving linear system should be included in addition to conjugate gradient method for a better comparison.
4. The applicability of the proposed approach is limited:
1). How do the end-users control the trade-off between precision and computation time?
2). The authors mentioned that "Although the Conjugate Gradient (CG) solution scales as O(N2), which is asymptotically faster than matrix multiplication’s O(N2.8), the constant overhead of CG becomes significant for the small N that we consider here." What would be a good N for applying proposed methods in practice?

---

### Official Review · Reviewer_6KUh · 2024-06-11

**Rating:** 5
**Confidence:** 3

**Summary:**

The paper presents an innovative approach to solving systems of linear equations using linear Transformers. The authors propose encoding each equation as a separate token, leveraging the permutation-invariant nature of self-attention mechanisms. They introduce a novel re-parameterization technique that decouples the problem dimension from the model’s parameter count, enabling efficient handling of systems of varying sizes. Experimental results demonstrate competitive performance compared to classical methods such as Conjugate Gradient, particularly for smaller systems, and the potential for extrapolation to larger systems.

**Questions:**

1. Why choose Linear Transformer over Standard Transformer?

**Reasons To Accept:**

1. The approach of using Transformers for solving linear systems is novel.
2. The proposed re-parameterization technique is novel, allowing the model to handle different problem sizes efficiently.

**Reasons To Reject:**

1. The paper focuses on solving linear systems with positive definite and symmetric matrices, which is a specific and somewhat limited scope.
2. The paper focuses on empirical results, but a deeper theoretical analysis of the proposed method's computational complexity and convergence properties would add significant value.
3. The paper demonstrates the method's efficiency for smaller systems, but its performance and scalability for very large systems remain unclear. Additional experiments with larger systems would strengthen the claims.

---

### Meta-Review · Area_Chair_F8R5 · 2024-06-13

**Recommendation:** Accept (Poster)
**Confidence:** 4

**Metareview:**

This paper proposes a linear Transformer for solving linear systems equations. This idea and the proposed re-parameterization technique are novel according to Reviewers 6KUh and dVvN. The authors could further improve this paper by including a deeper theoretical analysis (Reviewer 6KUh), large-scale experiments (Reviewers 6KUh, dVvN), additional baseline comparisons (Reviewers tFdu, dVvN), additional evaluation metrics (Reviewer dVvN), and polish the writing (Reviewers tFdu, dVvN).

---

### Decision · Program_Chairs · 2024-06-13

Accept (Poster)